psychology/cognition

developmental prosopagnosia, face recognition, Twenty Item Prosopagnosia Index (PI20), Cambridge Face Memory Test

**Author for correspondence:**
Maria Tsantani
e-mail: m.tsantani@bbk.ac.uk

# The Twenty Item Prosopagnosia Index (PI20) provides meaningful evidence of face recognition impairment

## Maria Tsantani, Tim Vestner and Richard Cook

Department of Psychological Sciences, Birkbeck, University of London, London, UK

MT, 0000-0002-4118-0031; TV, 0000-0002-7839-0390;
RC, 0000-0003-2370-3086

The Twenty Item Prosopagnosia Index (PI20) is a self-report questionnaire used for quantifying prosopagnosic traits. This scale is intended to help researchers identify cases of developmental prosopagnosia by providing standardized self-report evidence to complement diagnostic evidence obtained from objective computer-based tasks. In order to respond appropriately to items, prosopagnosics must have some insight that their face recognition is well below average, while non-prosopagnosics need to understand that their relative face recognition ability falls within the typical range. There has been considerable debate about whether participants have the necessary insight into their face recognition abilities to respond appropriately. In the present study, we sought to determine whether the PI20 provides meaningful evidence of face recognition impairment. In keeping with the intended use of the instrument, we used PI20 scores to identify two groups: high-PI20 scorers (those with self-reported face recognition difficulties) and low-PI20 scorers (those with no self-reported face recognition difficulties). We found that participant groups distinguished on the basis of PI20 scores clearly differed in terms of their mean performance on objective measures of face recognition ability. We also found that high-PI20 scorers were more likely to achieve levels of face recognition accuracy associated with developmental prosopagnosia.

## 1. Introduction

Face recognition ability varies substantially within the general population [1]. While some individuals (so-called 'super-recognizers') never forget a face [2], others are demonstrably

**Table 1.** The statements comprising the PI20.

1. My face recognition ability is worse than most people

2. I have always had a bad memory for faces

3. I find it notably easier to recognize people who have distinctive facial features

4. I often mistake people I have met before for strangers

5. When I was at school, I struggled to recognize my classmates

6. When people change their hairstyle or wear hats, I have problems recognizing them

7. I sometimes have to warn new people I meet that I am 'bad with faces'

8. I find it easy to picture individual faces in my mind*

9. I am better than most people at putting a 'name to a face'*

10. Without hearing people's voices, I struggle to recognize them

11. Anxiety about face recognition has led me to avoid certain social or professional situations

12. I have to try harder than other people to memorize faces

13. I am very confident in my ability to recognize myself in photographs*

14. I sometimes find movies hard to follow because of difficulties recognizing characters

15. My friends and family think I have bad face recognition or bad face memory

16. I feel like I frequently offend people by not recognizing who they are

17. It is easy for me to recognize individuals in situations that require people to wear similar clothes (e.g. suits, uniforms and swimwear)*

18. At family gatherings, I sometimes confuse individual family members

19. I find it easy to recognize celebrities in 'before-they-were-famous' photos, even if they have changed considerably*

20. It is hard to recognize familiar people when I meet them out of context (e.g. meeting a work colleague unexpectedly while shopping)

N.B. Asterisks indicate items that are reverse scored.

'bad with faces' [3]. Individuals with developmental prosopagnosia (DP) experience lifelong face recognition problems that present despite normal vision and cognitive abilities, and in the absence of brain damage [4–6]. DP affects the identification of various face types suggesting that it is distinguishable from poor face recognition arising from a lack of experience with other-ethnicity faces [7]. Although DPs struggle to recognize individuals from facial cues, they show typical person identification from vocal cues [8,9]. Historically, the condition was thought to be rare [10]. However, more recent estimates suggest that approximately 2% of the general population experience lifelong face recognition problems severe enough to disrupt their daily lives [11–13]. Evidence that the condition runs in families suggests that it may have a genetic component [14–18].

A diagnosis of DP is typically based on a combination of self-report evidence and poor performance on objective measures of face recognition, such as the Cambridge Face Memory Test (CFMT [19]) and the Cambridge Face Perception Test (CFPT [14]). Where observed, self-reported face recognition difficulties provide reassurance that poor performance on objective face recognition tests is due to DP, rather than poor motivation, fatigue or misinterpretation of the instructions [20,21]. Until recently, however, there was little attempt to quantify and standardize self-report evidence [12,22,23]. Reliance on bespoke interviews (e.g. [24]) or bespoke questionnaires (e.g. [25]), hindered the comparison of self-report evidence across different studies.

In 2015, Shah *et al*. [23] published the PI20, a self-report questionnaire for quantifying prosopagnosic traits. The PI20 is intended to help researchers diagnose cases of DP by providing standardized self-report evidence to complement diagnostic evidence obtained from objective computer-based tasks. The scale comprises 20 statements describing face recognition experiences drawn from qualitative and quantitative descriptions of DP (table 1). Respondents indicate to what extent the statements describe them on a five-point scale. Scores can range from 20 to 100. Shah *et al*. [23] suggested that a score of 65 or more may be suggestive of DP. Since its publication, the PI20 has been used in numerous research studies (e.g. [7,8,26–31]).

The items on the PI20 ask participants to assess their face recognition ability relative to the rest of the population, either explicitly (e.g. My face recognition ability is worse than most people; I am better than most people at putting a 'name to a face'; I have to try harder than other people to memorize faces) or implicitly (e.g. When people change their hairstyle or wear hats, I have problems recognizing them; when I was at school, I struggled to recognize my classmates). In order to respond appropriately to these items, prosopagnosics must have some insight that their face recognition is well below average, while non-prosopagnosics need to understand that their ability falls in the middle or upper tail of the distribution. There has been considerable debate about whether participants have this level of insight into their face recognition abilities [4,11,32–36].

If the PI20 produces meaningful evidence of face recognition impairment, respondents' scores should predict their performance on objective measures of face recognition ability (e.g. the CFMT). The authors of the PI20 presented evidence in support of this view. In a sample of 173 individuals (100 typical participants, 73 suspected DPs), Shah *et al.* [23] found a negative correlation of $r = -0.813$ between PI20 scores (higher scores indicate worse perceived ability) and famous face recognition. Similarly, in a sample of 110 individuals (87 typical participants, 23 suspected DPs), they found a correlation of $r = -0.683$ between PI20 scores and scores on the CFMT. Note that these samples are not typical of the wider population—the incidence of DP is thought to be approximately 2% [11–13]; rather, these analyses were intended to demonstrate that objective recognition ability varies as a function of PI20 score.

By contrast, several research groups have found that large samples drawn from the general population exhibit only modest insight into their face recognition ability [4,11,32–35]. For example, having asked undergraduates to rate their ability to recognize faces in everyday life 'compared with the average person', Bowles *et al.* [11] found only weak correlations between scores on this single-item measure and performance on the CFMT ($r = 0.220$) and CFPT ($r = -0.119$). Similarly, Matsuyoshi and Watanabe [33] found a correlation of $r = -0.23$ between scores on a Japanese translation of the PI20 and scores on an East Asian version of the CFMT. In the light of evidence that typical participants appear to have limited insight into their face recognition ability, some have argued that self-report evidence—notably scores on the PI20—have limited utility when identifying cases of DP [4,11,32–35].

In our view, these critiques of the PI20 are unfounded. In order to complete the PI20 appropriately, respondents require a very crude level of insight; whether they have difficulties with face recognition or not. It is misleading to argue that because typical controls do not possess more fine-grained knowledge— the ability to identify accurately where they fall within the typical range (e.g. 45% versus 55%)—that DPs and controls are unable to correctly identify as impaired or unimpaired. Very few situations arise that reveal to a member of the public whether their face recognition ability is slightly above average or slightly below average. By contrast, DP is socially debilitating and individuals with the condition frequently encounter embarrassing situations that reveal that they are less able to recognize familiar others from facial cues, than their colleagues, friends and family.

By way of analogy, a teacher, a bookkeeper and a librarian, may have only modest insight into the relative size of their respective salaries; however, this doesn't mean people are unaware whether or not they live in poverty. Here again, relatively few day-to-day experiences reveal to individuals whether their standard of living is slightly above or slightly below their national average. However, situations do arise that reveal whether you live in poverty: do you struggle to put food on the table? Can you afford to buy new clothes for your family? Most people, therefore, have a pretty good idea on which side of the poverty line they fall.

A second misconception is that the PI20 is a measure of face recognition ability, rather than a diagnostic instrument for identifying cases of DP [21]. Importantly, the instrument was designed to identify individuals with DP and to distinguish those with lifelong face recognition problems from everyone else. Non-DP participants will probably respond to many items (e.g. anxiety about face recognition has led me to avoid certain social or professional situations; I am very confident in my ability to recognize myself in photographs) in the same way, irrespective of whether they fall within the typical or superior range. The instrument, therefore, has little ability to distinguish people who are slightly below average, from those slightly above average, from super-recognizers. Evidence that the PI20 is a poor measure of typical participants' face recognition ability does not mean that the PI20 is ineffective at distinguishing likely DPs from likely non-DPs (i.e. its intended purpose).

In the present study, we revisit the question of whether suspected DPs and controls have sufficient insight into their relative face recognition abilities to render diagnostic information obtained from the PI20 meaningful. The value of the PI20 lies in its ability to identify potential DPs, not in its ability to describe face recognition variability in the general population. In keeping with its intended use, we adopt a group design (not a correlational approach). We use the PI20 to identify two groups of

participants: suspected DPs (high scorers) and suspected non-DPs (low scorers). Having defined groups of participants based solely on the individuals' PI20 scores, we examine how these groups differ in their performance on objective measures of face recognition ability (two variants of the CFMT). If PI20 scores provide meaningful diagnostic evidence, suspected DPs (identified by the PI20) should perform a lot worse than suspected non-DPs (identified by the PI20). Finally, we assess the ability of the PI20 to discriminate between CFMT-diagnosed DPs and typical participants using receiver operating characteristic (ROC) curves.

# 2. Methods

## 2.1. Participants

Three hundred and eighty-four participants (144 males, 1 non-binary, 2 undisclosed; $M_{age} = 36.8$ years, $s.d._{age} = 11.34$ years) were included in this study. All participants were between 21 and 60 years of age. Of this combined sample, 238 (104 males, 1 non-binary, 2 undisclosed; $M_{age} = 36.56$ years, $s.d._{age} = 11.72$ years) were recruited through Prolific (www.prolific.co). Participants reported being resident in the UK and had a minimum Prolific approval rate of 90%. We excluded participants with a diagnosis of autism spectrum disorder or other mental health conditions.

We anticipated that relatively few participants recruited through Prolific would score in the DP range on the PI20. We, therefore, recruited a further 146 participants (40 males; $M_{age} = 37.18$ years, $s.d._{age} = 10.73$ years) through www.troublewithfaces.org. On this website, individuals who experience face recognition difficulties register their interest to undergo further testing for prosopagnosia at a UK university. Potential participants were excluded if they described co-occurring psychiatric conditions (e.g. autism or schizophrenia), a history of brain damage or symptoms indicative of a stroke. Of the 146 participants recruited through troublewithfaces.org, 120 were tested online and 26 were tested in the laboratory.

## 2.2. Procedure

All participants first completed the PI20 [23], followed by the original version of the CFMT [19], and finally the Australian version (CFMT-A [37]). The online versions of the CFMT and CFMT-A were programmed in Unity (https://unity.com) and made available through WebGL. Participants completed the experiment on a desktop PC or laptop and recorded their responses using a keyboard. Data and an R analysis script are available via the Open Science Framework (https://osf.io/7a2u8/).

The CFMT and CFMT-A have an identical three-alternative-forced-choice match-to-sample format. On each trial, participants are required to identify a target face from a line-up containing the target and two foils. In the first task block, to-be-memorized faces are shown from different viewpoints. At test, participants are shown similar images and are asked to identify the target. In the second block, participants are required to memorize six target faces presented simultaneously. At test, participants are asked to identify novel images of the targets. The third block has the same format as the second block, but the image quality is degraded through the addition of visual noise. In total, there are 72 trials. Both the CFMT and the CFMT-A present Caucasian faces only.

## 2.3. Statistical procedures

Pairwise contrasts were calculated using Student's $t$-tests. Where comparisons were made between groups with unequal variance, we used Welch's $t$-test instead. We report Cohen's $d$ as a measure of effect size, calculated using the rstatix package (v. 0.7.0; function 'cohens_d'; [38]) in R (v. 4.0.4 [39]). We used chi-square tests to test associations between categorical variables and report the odds ratio as a measure of effect size. Correction for multiple comparisons was performed using the false discovery rate (FDR) with $q < 0.05$. Correlations were computed using Spearman's rank correlation. All reported $p$-values are two-tailed.

We used ROC curves to assess the ability of the PI20 to discriminate between CFMT-diagnosed DPs and typical participants. ROC curves and the area under the curve (AUC) were computed using the pROC package (v. 1.17.0.1; function 'roc'; [40]) in R. ROC curves show the sensitivity and specificity at each PI20 cut-off point, with sensitivity indicating the proportion of correctly identified DP participants (true positives) and specificity indicating the proportion of correctly identified typical

**Table 2.** Group performance on the PI20, CFMT and CFMT-A.

|  |  | age | PI20 | CFMT (%) | CFMT-A (%) |
|---|---|---|---|---|---|
| low-PI20 | mean | 36.14 | 43.43 | 74.55 | 76.10 |
| N = 225 | s.d. | 11.66 | 9.08 | 13.56 | 11.90 |
|  | range | 21–60 | 23–64 | 40.28–95.83 | 48.61–98.61 |
| high-PI20 | mean | 37.73 | 77.04 | 60.46 | 65 |
| N = 159 | s.d. | 10.85 | 7.49 | 12.38 | 13.47 |
|  | range | 21–60 | 65–96 | 31.94–93.06 | 36.11–97.22 |

participants (true negatives). The AUC provides an estimate of the probability that a randomly chosen DP will be distinguished from a randomly chosen typical participant based on the ordering of their PI20 scores [41]. An AUC value of 0.5 indicates no discriminatory ability, and 1 indicates perfect accuracy [41]. Confidence intervals for the AUC were obtained using DeLong's method [42]. Statistical significance for the AUC was computed used Wilcoxon rank sum tests [41]. Optimal cut-offs for the PI20 were obtained using Youden's J statistic [43] (function 'coords') and reflect the best combination of sensitivity and specificity.

# 3. Results

## 3.1. Comparing high and low scorers

For the purpose of the analysis, participants were divided into two groups based on their PI20 scores. Anyone who scored 65 or more was treated as a high scorer. This was the threshold originally identified by Shah *et al.* [23] as suggestive of DP. A low score was defined as 64 or below, a range of scores potentially indicative of typical face recognition ability.

The low-PI20 group ($N = 225$, 97 males, 1 non-binary, 1 undisclosed; $M_{age} = 36.14$ years, s.d.$_{age} = 11.66$ years) included participants recruited from Prolific who scored between 20 and 64. The high-PI20 group ($N = 159$; 47 males, 1 undisclosed; $M_{age} = 37.73$ years, s.d.$_{age} = 10.85$ years) included all 146 participants recruited through www.troublewithfaces.org and 13 participants recruited through Prolific who scored 65 or more on the PI20. As expected, the two groups differed significantly in terms of PI20 scores [$t_{372.969} = 39.642$, $p < 0.001$, $d = 4.039$], but did not differ significantly in terms of age [$t_{382} = 1.356$, $p = 0.176$, $d = 0.140$] (table 2).

As predicted, the high-PI20 group achieved lower mean accuracy scores on the CFMT (60.46%) and the CFMT-A (65%), compared with the low-PI20 group (CFMT: 74.55%, CFMT-A: 76.10%) (table 2; figure 1a). ANOVA with Group (low-PI20, high-PI20) as a between-subjects factor and CFMT version (CFMT, CFMT-A) as a within-subjects factor revealed a main effect of Group [$F_{1382} = 109.078$, $p < 0.001$, $\eta_p^2 = 0.222$]. Pairwise contrasts showed that the high-PI20 group achieved significantly lower accuracy scores than the low-PI20 group on both the CFMT [$t_{358.030} = 10.556$, $p < 0.001$, $d = 1.085$] and the CFMT-A [$t_{382} = 8.523$, $p < 0.001$, $d = 0.883$]. There was a main effect of CFMT version [$F_{1382} = 29.546$, $p < 0.001$, $\eta_p^2 = 0.072$]. Both the low-PI20 group [$t_{224} = 2.170$, $p = 0.031$, $d = 0.145$] and the high-PI20 group [$t_{158} = 5.228$, $p < 0.001$, $d = 0.415$] achieved higher scores on the CFMT-A, than on the CFMT. There was also a significant CFMT version × Group interaction [$F_{1382} = 7.114$, $p = 0.008$, $\eta_p^2 = 0.018$], whereby the effect of Group (low-PI20, high-PI20) was more pronounced on the CFMT, than on the CFMT-A. Because the CFMT-A was always completed after the CFMT, the main effect of CFMT version and the interaction with Group may reflect practice [44].

The high-PI20 group contained a higher proportion of participants who scored below 65%, 60% and 55% on the CFMT, on the CFMT-A and on both tests, compared with the low-PI20 group (table 3; figure 1b). Scores within this range are thought to be indicative of DP [9,17,35]. There were significant associations between Group (high-PI20 and low-PI20) and the proportion of participants who scored below each cut-off on the CFMT [65%: $\chi_{(1)}^2 = 55.396$, 60%: $\chi_{(1)}^2 = 52.354$, 55%: 40.940; all $p < 0.001$], on the CFMT-A [65%: $\chi_{(1)}^2 = 38.994$, 60%: $\chi_{(1)}^2 = 38.126$, 55%: 25.517; all $p < 0.001$] and on both tests [65%: $\chi_{(1)}^2 = 47.366$, 60%: $\chi_{(1)}^2 = 39.968$, 55%: 29.674; all $p < 0.001$], after controlling for the FDR. The odds of participants scoring below the cut-off were between 4.05 and 35.97 times higher if they were in the high-PI20 group, than if they were in the low-PI20 group (table 3).

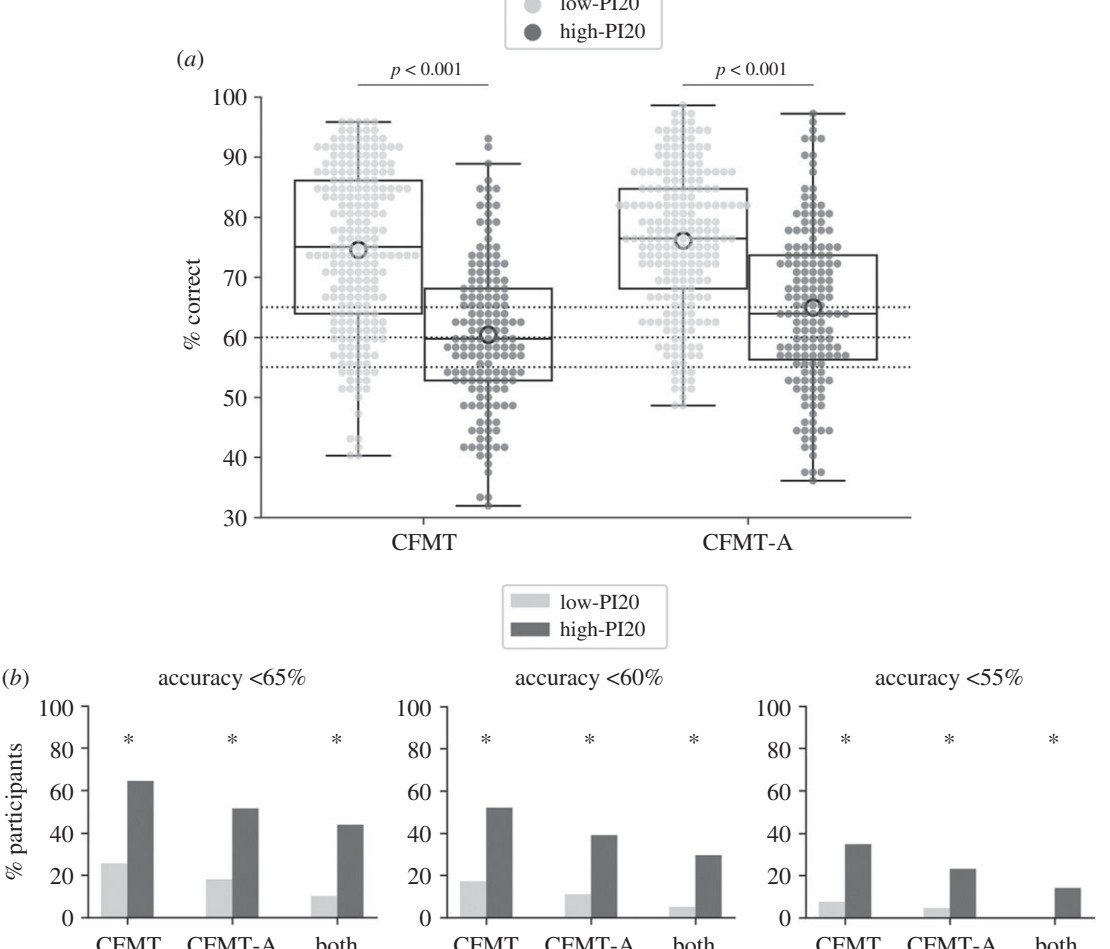

**Figure 1.** Summary of the results for the low-PI20 versus high-PI20 scorers. (*a*) Performance on the CFMT and CFMT-A. Boxes show the median and interquartile range, circles show the mean and dotted lines indicate the three CFMT cut-off scores used in the analysis: 65%, 60% and 55%. (*b*) Proportion of participants scoring below each cut-off in the CFMT, in the CFMT-A and in both tests. Asterisks indicate significant chi-square tests after controlling for the FDR ($p \leq 0.001$).

**Table 3.** Proportion of participants in each group that scored lower than 65%, 60% or 55% on the CFMT, on the CFMT-A, on both tests and the odds ratio for the high-PI20 group over the low-PI20 group.

|  |  | CFMT | CFMT-A | both |
|---|---|---|---|---|
| low-PI20 | <65% | 26.67% | 20% | 12% |
| $N = 225$ | <60% | 17.78% | 12% | 5.78% |
|  | <55% | 8.44% | 5.33% | 0.44% |
| high-PI20 | <65% | 64.78% | 50.31% | 42.77% |
| $N = 159$ | <60% | 52.83% | 38.99% | 29.56% |
|  | <55% | 34.59% | 22.64% | 13.84% |
| odds ratio | <65% | 5.06 | 4.05 | 5.48 |
|  | <60% | 5.18 | 4.69 | 6.84 |
|  | <55% | 5.73 | 5.20 | 35.97 |

For completeness, we also computed the correlations between PI20 scores and accuracy on the CFMT and CFMT-A in the low-PI20 scorers and in the high-PI20 scorers separately. For the low-PI20 scorers, correlations with the PI20 were $r_s = -0.23$ ($p < 0.001$) for both the CFMT and the CFMT-A. For the high-PI20 scorers, the correlations were $r_s = -0.21$ ($p = 0.009$) for the CFMT and $r_s = -0.36$ ($p < 0.001$)

**Table 4.** Results of the ROC analysis for each participant split showing the number of participants (*N*) who were classified as 'DP' or 'non-DP' based on the CFMT & CFMT-A, the AUC value and its 95% confidence interval (CI) and the optimal PI20 cut-off and associated sensitivity (% of DPs correctly identified) and specificity (% of non-DPs correctly identified).

| | | | AUC | | optimal PI20 cut-off | | |
|---|---|---|---|---|---|---|---|
| CFMT cut-off | *N* DPs | *N* non-DPs | value | CI | cut-off | sensitivity | specificity |
| 65% | 95 | 289 | 0.75 | 0.69–0.80 | 60.5 | 77% | 67% |
| 60% | 60 | 324 | 0.76 | 0.70–0.83 | 60.5 | 82% | 63% |
| 55% | 23 | 361 | 0.88 | 0.82–0.94 | 75.5 | 78% | 82% |

for the CFMT-A. Unsurprisingly, correlations were stronger when combining both groups: $r_s = -0.50$ ($p < 0.001$) for the CFMT and $r_s = -0.47$ ($p < 0.001$) for the CFMT-A.

## 3.2. Assessing the discrimination performance of the PI20

In our main analysis, we split participants into a group of low-PI20 scorers and a group of high-PI20 scorers based on a PI20 cut-off score of 65 and then analysed performance on the CFMT in each group. In this analysis, we reverse our approach by first 'diagnosing' our participants as 'DPs' or 'non-DPs' based on a given CFMT cut-off score, and then assessing the ability of the PI20 as a continuous measure to discriminate DPs from typical participants. Participants were classified as having DP if they scored lower than the cut-off on both versions of the CFMT (original version and CFMT-A). We used the same three CFMT cut-off scores (65%, 60% and 55%) employed in the previous analysis to create three separate splits of participants as DPs and non-DPs and then assessed the discrimination accuracy of the PI20 for each split. Table 4 shows the results of the ROC curve analyses. AUC values for the participant splits based on the 65%, 60% and 55% CFMT cut-offs were 0.75, 0.76 and 0.88, respectively (all $p < 0.001$). None of the confidence intervals crossed 0.5, confirming that the PI20 discriminates between DPs and non-DPs with better-than-chance accuracy.

The PI20 cut-off score with the optimal combination of sensitivity and specificity was 60.5 for the participant splits based on the 65% and 60% diagnostic cut-offs. For the participant split based on the stricter 55% cut-off, the optimal PI20 cut-off was 75.5. Sensitivity at these cut-offs ranged between 77% and 82% and specificity ranged between 63% and 82% (table 4).

## 4. General discussion

In the present study, we sought to determine whether high scores on the PI20 provide meaningful evidence of face recognition impairment. In keeping with the intended use of the instrument, we used PI20 scores to identify two groups: a group of high-PI20 scorers (suspected DPs) and a group of low-PI20 scorers (suspected non-DPs). We found that participant groups distinguished solely on the basis of PI20 scores differed in terms of their performance on the CFMT, one of the most commonly used objective measures of face recognition ability. The group of high-PI20 scorers ($N = 159$) achieved significantly lower scores on the original CFMT and on the CFMT-A, compared with the group of low-PI20 scorers ($N = 225$).

We also found that the high-scoring PI20 group contained a larger proportion of participants with low CFMT accuracy scores than the low-scoring PI20 group, for three different CFMT cut-off scores. Around 43% of the participants in the high-PI20 group scored lower than 65% on both the CFMT and CFMT-A, compared with 12% of the low-PI20 group. Even clearer differences between low-PI20 and high-PI20 scorers were observed when using the stricter cut-offs. Around 30% of the participants in the high-PI20 group scored lower than 60% on both the CFMT and CFMT-A, compared with only approximately 6% of the low-PI20 group. Similarly, approximately 14% of high-PI20 scorers scored below the 55% cut-off on both tests, while fewer than 1% of the low-PI20 scorers met this threshold.

These findings accord well with the results of a recent study that investigated the ability of the PI20 to identify members of the general population who score poorly on the original version of the CFMT [36]. Out of their sample ($N = 425$), Arizpe *et al.* found that 12 scored in the DP range on the PI20, and five (approx. 42%) of these participants scored lower than a 61% cut-off on the CFMT. Of the remaining

413 participants with low-PI20 scores, only 54 (approx. 13%) scored lower than 61% on the CFMT. Unlike Arizpe *et al.*, we were able to test a large sample of high-PI20 scorers ($N = 160$). Our results further confirm that participants who report poor face recognition ability via the PI20 are much more likely to achieve low CFMT scores, compared with participants who report good face recognition ability.

When we assessed the ability of the PI20 as a continuous measure to discriminate CFMT-diagnosed DPs from typical participants, we found that it achieved better-than-chance accuracy. Depending on the CFMT cut-off score that was used to classify participants as DPs or non-DPs, the probability of a randomly chosen DP being distinguished from a randomly chosen typical participant based on their PI20 scores ranged from 75% (for the most liberal cut-off) to 88% (for the strictest cut-off). Values between 70 and 80% are considered acceptable, values between 80 and 90% are considered excellent, and values of 90% and higher are considered exceptional [45].

We also identified the PI20 cut-off points with the optimal combinations of sensitivity and specificity as 60.5 for participant splits based on the upper (65%) and mid (60%) CFMT cut-offs, and 75.5 for the split based on the strictest (55%) CFMT cut-off. Shah *et al.* [23] originally recommended a cut-off of 65 to distinguish suspected DPs from suspected non-DPs. The present results suggest that scores of 61 or higher can be used to identify suspected DPs. We note that in our main analysis, the group of low-PI20 scorers contained nine participants who scored in the 61–64 range on the PI20. Of these nine participants, five scored below 65% on the CFMT, six scored below 65% on the CFMT-A, and five scored below 65% on both versions. In the existing literature, there is already some suggestion that scores in this range are a little ambiguous; for example, DPs are occasionally included in research samples with PI20 scores in this range [27].

Together, these results confirm that respondents have sufficient insight into their relative face recognition ability to respond appropriately to the items on the PI20. There is increasing evidence that individuals drawn from the general population have relatively poor insight into their relative face recognition ability; i.e. where they fall within the unimpaired range [4,11,32–35]. However, while participants may lack fine-grained knowledge about their relative face recognition ability, they seem to have a good idea whether or not they fall within the impaired or unimpaired range. While this level of insight may be relatively crude by comparison, it suggests that self-report instruments like the PI20 can play a useful role in identifying cases of DP.

Several studies have examined the relationship between PI20 and CFMT scores across the typical distribution [21,33,36]. These studies have found only modest correlations between PI20 scores and CFMT performance. In our data, we also found low correlations between PI20 and CFMT scores in our group of low-PI20 scorers and in our group high-PI20 scorers. Critically, the PI20 is a measure of prosopagnosic traits, not of face recognition *per se*. The scale has little ability to distinguish people who are slightly below average, from those slightly above average, from super-recognizers. As such, the modest correlations described above are uninformative about the validity of the scale. Insofar as correlational approaches assume a linear relationship between PI20 score and CFMT performance across the entire range of abilities, they are ill-suited to the validation of the PI20. The group design used here—in particular, the categorical treatment of anyone who scores below cut-off as unimpaired—provides a fairer test of the validity of this instrument.

To be clear, we are not arguing that diagnostic decisions should be based solely on self-report measures such as the PI20. Rather, we suggest that the use of convergent objective and self-report measures may afford the most reliable diagnosis of DP. People can over-perform on the CFMT due to practice effects (an increasing number of websites host versions of this test [44]), an above-average number of chance hits or because they are able to infer the correct solution by detecting trivial image-specific details (e.g. a specular highlight). Conversely, some individuals may underperform because they lack motivation, are fatigued, are unfamiliar with cognitive testing or mistakenly prioritize response speed over response accuracy. The convergent use of self-report evidence (e.g. PI20 scores) reduces the likelihood that genuinely impaired participants are excluded from DP research or that unimpaired participants are erroneously included in DP samples.

Online testing has been a great innovation that can help researchers achieve larger sample sizes. Evidence suggests that this approach can produce high-quality data comparable to that obtained from in-person testing [46–48]. To give recent examples from our own research, we have found that online testing has produced clear, replicable results in visual search [49,50] and attention cueing [51,52] experiments, and studies of visual illusions [53,54]. However, we acknowledge that this approach is associated with some well-known limitations. For example, it is not easy to control the testing environment, the participants' viewing distance or their monitor settings. For these reasons, we were unable to verify the low-level visual capacities of participants.

Previous research has confirmed that poor visual acuity can impede performance on face matching tasks [55]. Without data on participants' low-level visual abilities, we cannot rule out the possibility that some individuals who scored poorly on the CFMT had diminished visual acuity. The contribution of low-level visual problems to PI20 responses and poor face recognition are certainly important avenues for future research. However, previous findings suggest that low-level visual deficits are unlikely to be responsible for the association between PI20 scores and CFMT performance seen here. If the PI20 captured variability in visual acuity, one would expect PI20 scores to be a good predictor of performance on non-face object recognition tasks. However, this is not the case [23,56,57]. Participants' responses on the PI20 may be informed by a lifetime of experience (e.g. When I was at school, I struggled to recognize my classmates; It is easy for me to recognize individuals in situations that require people to wear similar clothes; It is hard for me to recognize familiar people when I meet them out of context) and are, therefore, relatively robust to recent deterioration in visual acuity.

In order to obtain a large enough sample, it was necessary to recruit the bulk of the high-PI20 scorers from a specialist website (troublewithfaces.org). Typically, the individuals who approach troublewithfaces.org (and other DP research groups such as faceblind.org) are keen to understand years of face recognition difficulties and embarrassment. They are motivated by a desire to learn about the condition and contribute to DP research. They frequently travel long distances with little or no compensation and are typically conscientious, committed participants. When we have been able to observe participants complete our tests in the laboratory, the effort they expend on each trial is obvious. By contrast, all of the low-PI20 scorers were recruited via Prolific. It is conceivable that these individuals were less motivated to perform well. It is noteworthy that the mean CFMT performance of the low-PI20 scorers was 74.55%, a little below the typical mean of 80.4% originally described by Duchaine and Nakayama [19]. Had the Prolific participants been as motivated as those recruited through troublewithfaces.org, the relationship between PI20 and CFMT might have been even more striking.

In conclusion, the present work demonstrates that the PI20 can play a useful role in the identification of individuals with DP. Grouping participants as suspected DPs and non-DPs based on their PI20 scores revealed that the suspected DP group had significantly poorer face recognition ability and contained a greater proportion of individuals achieving low-accuracy scores on the CFMT, compared with suspected typical participants. Furthermore, we found that the PI20 could discriminate CFMT-diagnosed DPs from typical participants with better-than-chance accuracy (between 75% and 88% depending on the cut-off employed). Taken together, these findings suggest that individuals with DP tend to be aware that their face recognition ability is outside of the normal range, and this awareness is reflected in their responses on the PI20. While DP diagnoses should not be based solely on self-report measures, our findings support the use of the PI20 as a screening tool for researchers who are interested in recruiting individuals with DP.

Ethics. The study was approved by the Departmental Ethics Committee for Psychological Sciences, Birkbeck, University of London, and was conducted in accordance with the ethical guidelines laid down in the 6th (2008) Declaration of Helsinki. Participants provided informed consent.

Data accessibility. Data and an R analysis script are available via the Open Science Framework [58] (https://osf.io/7a2u8/).

Authors' contributions. R.C. conceived of the study. M.T. and R.C. designed the study. T.V. programmed the online experimental tasks. M.T. and T.V. collected the data. M.T. analysed the data. M.T. and R.C. wrote the manuscript. All authors gave approval for publication and agree to be held accountable for the work performed therein.

Competing interests. We declare we have no competing interests.

Funding. R.C. is supported by a Starting Grant awarded by the European Research Council (grant no. ERC-2016-StG-715824).

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
