## [Peer Review File · Royal Society Open Science]

Review History

RSOS-202062.R0 (Original submission)

Review form: Reviewer 1

Is the manuscript scientifically sound in its present form?

Yes

Are the interpretations and conclusions justified by the results?

No

Is the language acceptable?

Yes

Do you have any ethical concerns with this paper?

No

Have you any concerns about statistical analyses in this paper?

Yes

Recommendation?

Major revision is needed (please make suggestions in comments)

Comments to the Author(s)

Tsantani and colleagues present an interesting and timely study about the usefulness of subjective face recognition reports, in particular the PI20, in identifying individuals with objective face recognition abilities. The PI20 has started to become widely adopted by prosopagnosia researchers and I think the manuscript contributes to the debate about whether members of the general population and prosopagnosics have insight into their face recognition abilities. That said, there are some major issues that need to be addressed before recommending publication.

The first issue is that in the group comparisons (PI20 > 60 vs. 60 or lower) are confounded with how the two samples in the study were recruited, which may be potentially driving some of the differences between the groups. For the prolific sample that provides most of the 'PI20 60 or lower' group, this is more of a general community sample who are coming to the website because they are interested in making money and/or learning about their cognitive abilities. They are also motivated to perform well to keep up their prolific approval rate. In contrast, individuals coming to troublewithfaces.org (who mostly make up the PI20 > 60 group) are likely there because they suspect they have face recognition difficulties. This could potentially bias their self-report questionnaires such as the PI20 and perhaps even their objective test results. Further, individuals going to troublewithfaces.org may also generally have more issues such as undiagnosed autism (it doesn't sound like the authors administered the autism quotient questionnaire), undiagnosed developmental issues/brain injuries, and other potential general visual or memory issues. With such different recruitment for the two sample, I feel like comparisons within each sample would be much more valid.

The other issue is that the cutoff of 60 on the PI20 seems rather arbitrary. Previously it was reported that 65 was a reasonable cutoff to identify prosopagnosics but studies have included participants in prosopagnosia groups with scores of 59. Perhaps a better way to establish a PI20 cutoff is to perform ROC analyses for a particular target population and figure out a reasonable criterion for sensitivity and specificity. From Arizpe et al's recent findings, it doesn't seem like the general population is where the PI20 is most appropriate. It seems like the most useful place for the PI20 is to apply to all individuals that "think they may have prosopagnosia/severe face recognition problems" (or maybe all individuals that visit troublewithfaces.org). Then you could establish a cutoff that would reasonably discriminate those that actually have gold-standard verified prosopagnosia (impaired on at least two face recognition measures) vs. those who do not. These cutoff scores would be quite useful to prosopagnosia researchers who often get contacted by people with face recognition problems and they want to determine if they likely to be severe enough to include in their studies.

Review form: Reviewer 2

Is the manuscript scientifically sound in its present form?

Yes

Are the interpretations and conclusions justified by the results?

Yes

Is the language acceptable?

Yes

Do you have any ethical concerns with this paper?

No

Have you any concerns about statistical analyses in this paper?

No

Recommendation?

Accept with minor revision (please list in comments)

Comments to the Author(s)

The research provides a valuable contribution to the literature. It examines the meaningfulness of the PI20 as a tool to identify people who are likely to have Developmental Prosopagnosia. It uses a novel methodology to establish that participants who score high on the PI20 also perform worse on two versions of the CFMT, a widely used objective measure of face recognition ability, than participants who had low scores on the PI20. This indicates that the PI20 could be a useful tool to support screening of potential developmental prosopagnosics. This manuscript will be of particular interest to other researchers in the field who conduct research on DP as identification of larger DP samples is essential to progress our understanding of this developmental disorder. However there are a number of issues that I would hope the authors would address in a revised version of the manuscript;

Page 4, Line 51-52: can the authors please provide some explanation as to why DP samples have since included participants with scores below the cut-off proposed by Shah et al., (2015) as currently this reads rather contradictory that a cut off was established when the tool was initially validated and since ignored. What rationale was provided within those papers for including participants not reaching the threshold of 65?

Page 5, Line 16 and 17: the authors comment that there has been considerable debate about whether participants have this level of insight into their face recognition abilities – is there any published evidence of this debate that authors could refer readers to for a more detailed discussion of this issue? If not can the authors add a brief summary of this debate. I believe many of the points debated have been included in the introduction section but discussion of these points is rather brief so if readers cannot be directed to a more in depth discussion on this issue the paper would benefit from further detailed discussion of some of these points of debate.

Method: were any procedures put in place to ensure the reliability of the online data – beyond ensuring those recruited via prolific had a score of 90%+? (e.g. to ensure participants completed the CFMT tasks individually). If so the MS would benefit from these techniques being detailed to enhance the readers trust in the data.

Page 8, line 29: Can the authors please provide a rationale for using a cut-off point of 61 rather than the 65 originally proposed by Shah et al., (2015)? Some rationale is provided based on the fact other researchers have lowered the cut-off point, but the argument would be stronger with a scientific basis for lowering the cut-off point from that of the initial validation paper.

More generally, the aim of the paper is not entirely clear. The explicitly stated aim is to address the question of whether DPs and controls have sufficient insight in their face recognition ability for the PI20 to be meaningful, a question that I believe the manuscript sufficiently addresses. However, the introduction appears to aim to provide a defence of the PI20 as a tool to identify people with DP. The first aim is achieved but I am not convinced of the latter. If the intended use of the PI20 is to identify suspected DPs it seems unusual that the authors did not conduct a full battery assessment of the high scorers to establish whether their performance is consistent with the profile of DP. That is, yes the authors have found that high PI20 scorers perform worse on the

CFMT, a frequently used test that forms part of a typical assessment for DP. However, a poor score on CFMT alone or combined with a poor score on the CFMT-a is not sufficient to identify a person as having DP. Further tests / assessment is required to rule out alternative explanations of the face recognition difficulties (e.g. low-level visual difficulties, more general object recognition difficulties). These additional assessments would have better established the PI20's ability to discriminate between DPs and non-DPs. If the authors don't have data available to add this level of detail to the manuscript some consideration of this limitation of the current study is warranted. The findings provide convincing evidence that there is an association between PI20 scores and performance on CMFT and CMFTa as objective measures of face recognition but that doesn't necessarily mean an association with DP per se which I think needs to be considered.

Decision letter (RSOS-202062.R0)

Dear Dr Tsantani

The Editors assigned to your paper RSOS-202062 "The Twenty Item Prosopagnosia Index (PI20) provides meaningful evidence of face recognition impairment" have now received comments from reviewers and would like you to revise the paper in accordance with the reviewer comments and any comments from the Editors. Please note this decision does not guarantee eventual acceptance.

Please submit your revised manuscript and required files (see below) no later than 21 days from today's (ie 26-Mar-2021) date. Note: the ScholarOne system will 'lock' if submission of the revision is attempted 21 or more days after the deadline. If you do not think you will be able to meet this deadline please contact the editorial office immediately.

on behalf of Dr Bruno Rossion (Associate Editor) and Essi Viding (Subject Editor)
openscience@royalsociety.org

Reviewer comments to Author:

Reviewer: 1

Comments to the Author(s)

Tsantani and colleagues present an interesting and timely study about the usefulness of subjective face recognition reports, in particular the PI20, in identifying individuals with objective face recognition abilities. The PI20 has started to become widely adopted by prosopagnosia researchers and I think the manuscript contributes to the debate about whether members of the general population and prosopagnosics have insight into their face recognition abilities. That said, there are some major issues that need to be addressed before recommending publication.

The first issue is that in the group comparisons (PI20 > 60 vs. 60 or lower) are confounded with how the two samples in the study were recruited, which may be potentially driving some of the differences between the groups. For the prolific sample that provides most of the 'PI20 60 or lower' group, this is more of a general community sample who are coming to the website because they are interested in making money and/or learning about their cognitive abilities. They are also motivated to perform well to keep up their prolific approval rate. In contrast, individuals coming to troublewithfaces.org (who mostly make up the PI20 > 60 group) are likely there because they suspect they have face recognition difficulties. This could potentially bias their self-report questionnaires such as the PI20 and perhaps even their objective test results. Further, individuals going to troublewithfaces.org may also generally have more issues such as undiagnosed autism (it doesn't sound like the authors administered the autism quotient questionnaire), undiagnosed developmental issues/brain injuries, and other potential general visual or memory issues. With such different recruitment for the two sample, I feel like comparisons within each sample would be much more valid.

The other issue is that the cutoff of 60 on the PI20 seems rather arbitrary. Previously it was reported that 65 was a reasonable cutoff to identify prosopagnosics but studies have included participants in prosopagnosia groups with scores of 59. Perhaps a better way to establish a PI20 cutoff is to perform ROC analyses for a particular target population and figure out a reasonable criterion for sensitivity and specificity. From Arizpe et al's recent findings, it doesn't seem like the general population is where the PI20 is most appropriate. It seems like the most useful place for the PI20 is to apply to all individuals that "think they may have prosopagnosia/severe face recognition problems" (or maybe all individuals that visit troublewithfaces.org). Then you could establish a cutoff that would reasonably discriminate those that actually have gold-standard verified prosopagnosia (impaired on at least two face recognition measures) vs. those who do not. These cutoff scores would be quite useful to prosopagnosia researchers who often get contacted by people with face recognition problems and they want to determine if they likely to be severe enough to include in their studies.

Reviewer: 2

Comments to the Author(s)

The research provides a valuable contribution to the literature. It examines the meaningfulness of the PI20 as a tool to identify people who are likely to have Developmental Prosopagnosia. It uses

a novel methodology to establish that participants who score high on the PI20 also perform worse on two versions of the CFMT, a widely used objective measure of face recognition ability, than participants who had low scores on the PI20. This indicates that the PI20 could be a useful tool to support screening of potential developmental prosopagnosics. This manuscript will be of particular interest to other researchers in the field who conduct research on DP as identification of larger DP samples is essential to progress our understanding of this developmental disorder. However there are a number of issues that I would hope the authors would address in a revised version of the manuscript;

Page 4, Line 51-52: can the authors please provide some explanation as to why DP samples have since included participants with scores below the cut-off proposed by Shah et al., (2015) as currently this reads rather contradictory that a cut off was established when the tool was initially validated and since ignored. What rationale was provided within those papers for including participants not reaching the threshold of 65?

Page 5, Line 16 and 17: the authors comment that there has been considerable debate about whether participants have this level of insight into their face recognition abilities – is there any published evidence of this debate that authors could refer readers to for a more detailed discussion of this issue? If not can the authors add a brief summary of this debate. I believe many of the points debated have been included in the introduction section but discussion of these points is rather brief so if readers cannot be directed to a more in depth discussion on this issue the paper would benefit from further detailed discussion of some of these points of debate.

Method: were any procedures put in place to ensure the reliability of the online data – beyond ensuring those recruited via prolific had a score of 90%+? (e.g. to ensure participants completed the CFMT tasks individually). If so the MS would benefit from these techniques being detailed to enhance the readers trust in the data.

Page 8, line 29: Can the authors please provide a rationale for using a cut-off point of 61 rather than the 65 originally proposed by Shah et al., (2015)? Some rationale is provided based on the fact other researchers have lowered the cut-off point, but the argument would be stronger with a scientific basis for lowering the cut-off point from that of the initial validation paper.

More generally, the aim of the paper is not entirely clear. The explicitly stated aim is to address the question of whether DPs and controls have sufficient insight in their face recognition ability for the PI20 to be meaningful, a question that I believe the manuscript sufficiently addresses. However, the introduction appears to aim to provide a defence of the PI20 as a tool to identify people with DP. The first aim is achieved but I am not convinced of the latter. If the intended use of the PI20 is to identify suspected DPs it seems unusual that the authors did not conduct a full battery assessment of the high scorers to establish whether their performance is consistent with the profile of DP. That is, yes the authors have found that high PI20 scorers perform worse on the CFMT, a frequently used test that forms part of a typical assessment for DP. However, a poor score on CFMT alone or combined with a poor score on the CFMT-a is not sufficient to identify a person as having DP. Further tests / assessment is required to rule out alternative explanations of the face recognition difficulties (e.g. low-level visual difficulties, more general object recognition difficulties). These additional assessments would have better established the PI20's ability to discriminate between DPs and non-DPs. If the authors don't have data available to add this level of detail to the manuscript some consideration of this limitation of the current study is warranted. The findings provide convincing evidence that there is an association between PI20 scores and performance on CMFT and CMFTa as objective measures of face recognition but that doesn't necessarily mean an association with DP per se which I think needs to be considered.

===PREPARING YOUR MANUSCRIPT===

===PREPARING YOUR REVISION IN SCHOLARONE===

- An individual file of each figure (EPS or print-quality PDF preferred [either format should be produced directly from original creation package], or original software format).
- An editable file of each table (.doc, .docx, .xls, .xlsx, or .csv).
- An editable file of all figure and table captions.

- Any electronic supplementary material (ESM).
- If you are requesting a discretionary waiver for the article processing charge, the waiver form must be included at this step.
- If you are providing image files for potential cover images, please upload these at this step, and inform the editorial office you have done so. You must hold the copyright to any image provided.
- A copy of your point-by-point response to referees and Editors. This will expedite the preparation of your proof.

- Ensure that your data access statement meets the requirements at <https://royalsociety.org/journals/authors/author-guidelines/#data>. You should ensure that you cite the dataset in your reference list. If you have deposited data etc in the Dryad repository, please include both the 'For publication' link and 'For review' link at this stage.
- If you are requesting an article processing charge waiver, you must select the relevant waiver option (if requesting a discretionary waiver, the form should have been uploaded at Step 3 'File upload' above).
- If you have uploaded ESM files, please ensure you follow the guidance at <https://royalsociety.org/journals/authors/author-guidelines/#supplementary-material> to include a suitable title and informative caption. An example of appropriate titling and captioning may be found at https://figshare.com/articles/Table_S2_from_Is_there_a_trade-off_between_peak_performance_and_performance_breadth_across_temperatures_for_aerobic_scope_in_teleost_fishes_/3843624.

Author's Response to Decision Letter for (RSOS-202062.R0)

See Appendix A.

RSOS-202062.R1 (Revision)

Review form: Reviewer 1

Is the manuscript scientifically sound in its present form?

No

Are the interpretations and conclusions justified by the results?

No

Is the language acceptable?

Yes

Do you have any ethical concerns with this paper?

No

Have you any concerns about statistical analyses in this paper?

No

Recommendation?

Reject

Comments to the Author(s)

While I appreciate the authors' changes, the issue of having a confound that participants who comprised the low PI-20 group vs. high PI-20 were recruited from different sources remains unresolved. Maybe we just are learning that individuals who self-select and go to troublewithfaces.com are generally worse at face recognition than the general population of individuals at prolific. I'm not sure if more can be concluded from the results. Also, I would have preferred the authors perform an ROC analysis similar to Arizpe et al rather than go back to a relatively arbitrary cutoff score.

Review form: Reviewer 2

Is the manuscript scientifically sound in its present form?

Yes

Are the interpretations and conclusions justified by the results?

Yes

Is the language acceptable?

Yes

Do you have any ethical concerns with this paper?

No

Have you any concerns about statistical analyses in this paper?

No

Recommendation?

Accept with minor revision (please list in comments)

Comments to the Author(s)

This revised manuscript clearly addresses the major concerns raised by the reviewers, in particular the issue relating to the cut-off point for the high PI20 group. I welcome the revised analysis using the more conservative cut-off of 65. Whilst both the introduction and discussion sections remain brief I believe this article does provide a valuable contribution to the field, serving to illustrate the insight participants have into their own face recognition ability which further establishes the PI20 as a useful tool to support the identification of participants with suspected DP. I would therefore recommend acceptance of the article with just a couple of minor revisions;

Page 8 - the new paragraph says "this was the threshold originally identified Shah et al [23]" - I believe "by" is missing after identified within this sentence.

Page 10 - final sentence about low-level visual capacities. Could the authors please expand on this point to clearly explain why this is important for the readers to consider i.e. the potential impact it could have on the findings. It is now clear that participants were excluded if they had autism, schizophrenia or other mental health conditions etc. This additional information is the method is very welcome. However, it's not clear any attempt was made to check that poor performance on all measures wasn't due to differences in low-level visual perception / ability. It's possible someone with poor vision / low-level visual perceptual difficulties might score high on the PI20 and low on the CFMT but that the origins of their impairment be different to that of someone with DP who has scored high on the PI20 and low on the CFMT. So this point needs to be considered more fully not just within the context of online testing which is where it is currently raised.

Discussion - the authors might want to consider adding a conclusion paragraph at the end of the ms. The additions to the discussion are an improvement to the ms addressing some of the concerns raised by the reviewers of the original ms but they feel somewhat bolted on at the end of the ms and a return to the take home message, in spite of the limitations now acknowledged, would be beneficial to end the ms focussed on the important study aims.

Review form: Reviewer 3 (David White)

Is the manuscript scientifically sound in its present form?

No

Are the interpretations and conclusions justified by the results?

No

Is the language acceptable?

Yes

Do you have any ethical concerns with this paper?

No

Have you any concerns about statistical analyses in this paper?

Yes

Recommendation?

Major revision is needed (please make suggestions in comments)

Comments to the Author(s)

The paper presents a systematic evaluation of the diagnostic value of the PI-20 - a self-report measure of face recognition difficulties in daily life - in predicting scores on a test of relatively short-term unfamiliar face memory - the CFMT - which is a standard objective measure of face recognition ability. This is a useful study and method, presentation of results and reporting is generally high quality. The paper is well written

My major concern is the method for separating Low v High PI-20 scorers. The authors discard an undisclosed* proportion of participants with PI-20 scores between 61 and 64 , terming this data as 'ambiguous' . But I don't think it is acceptable to present this incomplete picture in a diagnostic

evaluation like this -- because the whole point of a diagnostic tool is that it allows researchers to set a criteria for diagnosis. It is perfectly OK to assess different criteria -- for example comparing the diagnostic value of a 60 v 65 PI-20 cutoff , but to exclude bands of data from the analysis is misleading and will lead to an inflated sense of the diagnostic value in researchers / clinicians that read the paper by casting their eye over the abstract and the main data figures.

This practice is equivalent to conducting a psychophysics experiment, where participants provide judgments of perceptual certainty on a likert scale, and removing data around the midpoint of the scale before calculating a participants' accuracy.

At the very least the CFMT data from this ambiguous group should be presented in the figures. But perhaps there is an opportunity to do something more holistic. For example, Area Under the ROC curve is a useful standard approach used in medical diagnosis (e.g. <https://www.sciencedirect.com/science/article/pii/S1556086415306043> , <https://www.ncbi.nlm.nih.gov/pmc/articles/PMC3755824/>). In the present study, this could be used by classifying the participants as CP / nonCP based in the CFMT, and using the continuous PI-20 score to predict this state (perhaps repeating this analysis for the different CFMT cutoffs in the current analysis).

MINOR

* the proportion of participants with PI-20 between 61 and 64 is undisclosed because the total number of participants reported to have completed the experiment do not include these participants. But given that all participants completed the PI-20, I am assuming that the full dataset must have included participants scoring in this range.

Figure 1 - (a) and (b) -- I think this plot would be clearer if y-axis was labelled 'Percent correct'. Also - could the two figures be combined simply by adding the cutoff lines in (b) to (a)? Please also clarify in the figure captions that the cutoffs relate to CFMT scores. I know it can be deduced from the information but worth making it abundantly clear.

=====
David White
UNSW SYDNEY

Review form: Reviewer 4

Is the manuscript scientifically sound in its present form?

Yes

Are the interpretations and conclusions justified by the results?

Yes

Is the language acceptable?

Yes

Do you have any ethical concerns with this paper?

No

Have you any concerns about statistical analyses in this paper?

No

Recommendation?

Accept with minor revision (please list in comments)

Comments to the Author(s)

Having read the authors' response to reviews as well as the revised text, I have found them to be responsive to the main issues raised in the last round of review. I agree that their justification of the cut-off values used to identify subgroups of suspected DP and non-DP participants are more clearly stated and I think that while recruiting from different outlets is not ideal, it also seems like it was likely necessary to ensure a large enough sample with sufficient variability to carry out their analysis. I think the only thing that could strengthen the main results somewhat is highlighting how the correlation between PI20 scores and CFMT performance plays out in both subgroups: I imagine poorly, which is largely their point! I think this could provide useful context given the conflicting results they describe in the introduction, but would not want to make this a condition for acceptance.

Otherwise, I think that the potential comorbidities of suspected DP with other aspects of object recognition and lower level visual processes could be given a little more space, but these also seem somewhat ancillary to the main point: The PI20 does indeed support the identification of participant groups with low vs. high face recognition performance as indexed by the CFMT. While more generality (perhaps examining instruments like the CFPT or the Glasgow Face Matching test) could be additionally useful, the paper accomplishes the narrower goal highlighted in the introduction.

Decision letter (RSOS-202062.R1)

Dear Dr Tsantani

The Editors assigned to your paper RSOS-202062.R1 "The Twenty Item Prosopagnosia Index (PI20) provides meaningful evidence of face recognition impairment" have now received comments from reviewers and would like you to revise the paper in accordance with the reviewer comments and any comments from the Editors. Please note this decision does not guarantee eventual acceptance.

Please submit your revised manuscript and required files (see below) no later than 21 days from today's (ie 03-Aug-2021) date. Note: the ScholarOne system will 'lock' if submission of the

revision is attempted 21 or more days after the deadline. If you do not think you will be able to meet this deadline please contact the editorial office immediately.

on behalf of Dr Bruno Rossion (Associate Editor) and Essi Viding (Subject Editor)
openscience@royalsociety.org

Associate Editor Comments to Author (Dr Bruno Rossion):

I apologize for the time to reach a decision, but following the first round of revision, one of the initial reviewer was not convinced by the modifications and recommended rejection of the paper. The other reviewer, while being positive overall, still had some concerns about the validity of the claims made in the revised paper. Two new reviewers, also experts in the field, were requested, and both recommend publication of the paper after revisions. However, reviewer 3 raises an important issue regarding data discarded from an undisclosed proportion of participants with PI-20 scores between 61 and 64, this data being defined by the authors as 'ambiguous'. This data should be included in the analysis and figures to avoid that diagnostic value of the test is artificially inflated. Please take into account all of the comments of the reviewers seriously, especially those of reviewer 3, for further revision of the paper.

Reviewer comments to Author:

Reviewer: 1

Comments to the Author(s)

While I appreciate the authors' changes, the issue of having a confound that participants who comprised the low PI-20 group vs. high PI-20 were recruited from different sources remains unresolved. Maybe we just are learning that individuals who self-select and go to troublewithfaces.com are generally worse at face recognition than the general population of individuals at prolific. I'm not sure if more can be concluded from the results. Also, I would have preferred the authors perform an ROC analysis similar to Arizpe et al rather than go back to a relatively arbitrary cutoff score.

Reviewer: 2

Comments to the Author(s)

This revised manuscript clearly addresses the major concerns raised by the reviewers, in particular the issue relating to the cut-off point for the high PI20 group. I welcome the revised analysis using the more conservative cut-off of 65. Whilst both the introduction and discussion sections remain brief I believe this article does provide a valuable contribution to the field, serving to illustrate the insight participants have into their own face recognition ability which further establishes the PI20 as a useful tool to support the identification of participants with

suspected DP. I would therefore recommend acceptance of the article with just a couple of minor revisions;

Page 8 - the new paragraph says "this was the threshold originally identified Shah et al [23]" - I believe "by" is missing after identified within this sentence.

Page 10 - final sentence about low-level visual capacities. Could the authors please expand on this point to clearly explain why this is important for the readers to consider i.e. the potential impact it could have on the findings. It is now clear that participants were excluded if they had autism, schizophrenia or other mental health conditions etc. This additional information is the method is very welcome. However, it's not clear any attempt was made to check that poor performance on all measures wasn't due to differences in low-level visual perception / ability. It's possible someone with poor vision / low-level visual perceptual difficulties might score high on the PI20 and low on the CFMT but that the origins of their impairment be different to that of someone with DP who has scored high on the PI20 and low on the CFMT. So this point needs to be considered more fully not just within the context of online testing which is where it is currently raised.

Discussion - the authors might want to consider adding a conclusion paragraph at the end of the ms. The additions to the discussion are an improvement to the ms addressing some of the concerns raised by the reviewers of the original ms but they feel somewhat bolted on at the end of the ms and a return to the take home message, in spite of the limitations now acknowledged, would be beneficial to end the ms focussed on the important study aims.

Reviewer: 3

Comments to the Author(s)

The paper presents a systematic evaluation of the diagnostic value of the PI-20 - a self-report measure of face recognition difficulties in daily life - in predicting scores on a test of relatively short-term unfamiliar face memory - the CFMT - which is a standard objective measure of face recognition ability. This is a useful study and method, presentation of results and reporting is generally high quality. The paper is well written

My major concern is the method for separating Low v High PI-20 scorers. The authors discard an undisclosed* proportion of participants with PI-20 scores between 61 and 64 , terming this data as 'ambiguous' . But I don't think it is acceptable to present this incomplete picture in a diagnostic evaluation like this -- because the whole point of a diagnostic tool is that it allows researchers to set a criteria for diagnosis. It is perfectly OK to assess different criteria -- for example comparing the diagnostic value of a 60 v 65 PI-20 cutoff , but to exclude bands of data from the analysis is misleading and will lead to an inflated sense of the diagnostic value in researchers / clinicians that read the paper by casting their eye over the abstract and the main data figures.

This practice is equivalent to conducting a psychophysics experiment, where participants provide judgments of perceptual certainty on a likert scale, and removing data around the midpoint of the scale before calculating a participants' accuracy.

At the very least the CFMT data from this ambiguous group should be presented in the figures. But perhaps there is an opportunity to do something more holistic. For example, Area Under the ROC curve is a useful standard approach used in medical diagnosis (e.g. <https://www.sciencedirect.com/science/article/pii/S1556086415306043> , <https://www.ncbi.nlm.nih.gov/pmc/articles/PMC3755824/>). In the present study, this could be used by classifying the participants as CP / nonCP based in the CFMT, and using the continuous PI-20 score to predict this state (perhaps repeating this analysis for the different CFMT cutoffs in the current analysis).

MINOR

* the proportion of participants with PI-20 between 61 and 64 is undisclosed because the total number of participants reported to have completed the experiment do not include these participants. But given that all participants completed the PI-20, I am assuming that the full dataset must have included participants scoring in this range.

Figure 1 - (a) and (b) -- I think this plot would be clearer if y-axis was labelled 'Percent correct'. Also - could the two figures be combined simply by adding the cutoff lines in (b) to (a)? Please also clarify in the figure captions that the cutoffs relate to CFMT scores. I know it can be deduced from the information but worth making it abundantly clear.

David White
UNSW SYDNEY

Reviewer: 4

Comments to the Author(s)

Having read the authors' response to reviews as well as the revised text, I have found them to be responsive to the main issues raised in the last round of review. I agree that their justification of the cut-off values used to identify subgroups of suspected DP and non-DP participants are more clearly stated and I think that while recruiting from different outlets is not ideal, it also seems like it was likely necessary to ensure a large enough sample with sufficient variability to carry out their analysis. I think the only thing that could strengthen the main results somewhat is highlighting how the correlation between PI20 scores and CFMT performance plays out in both subgroups: I imagine poorly, which is largely their point! I think this could provide useful context given the conflicting results they describe in the introduction, but would not want to make this a condition for acceptance.

Otherwise, I think that the potential comorbidities of suspected DP with other aspects of object recognition and lower level visual processes could be given a little more space, but these also seem somewhat ancillary to the main point: The PI20 does indeed support the identification of participant groups with low vs. high face recognition performance as indexed by the CFMT. While more generality (perhaps examining instruments like the CFPT or the Glasgow Face Matching test) could be additionally useful, the paper accomplishes the narrower goal highlighted in the introduction.

===PREPARING YOUR MANUSCRIPT===

Your revised paper should include the changes requested by the referees and Editors of your manuscript. You should provide two versions of this manuscript and both versions must be provided in an editable format:
one version identifying all the changes that have been made (for instance, in coloured highlight, in bold text, or tracked changes);
a 'clean' version of the new manuscript that incorporates the changes made, but does not highlight them. This version will be used for typesetting if your manuscript is accepted.

Please ensure that you include an acknowledgements' section before your reference list/bibliography. This should acknowledge anyone who assisted with your work, but does not

qualify as an author per the guidelines at <https://royalsociety.org/journals/ethics-policies/openness/>.

===PREPARING YOUR REVISION IN SCHOLARONE===

- Ensure that your data access statement meets the requirements at <https://royalsociety.org/journals/authors/author-guidelines/#data>. You should ensure that you cite the dataset in your reference list. If you have deposited data etc in the Dryad repository, please include both the 'For publication' link and 'For review' link at this stage.
- If you are requesting an article processing charge waiver, you must select the relevant waiver option (if requesting a discretionary waiver, the form should have been uploaded at Step 3 'File upload' above).
- If you have uploaded ESM files, please ensure you follow the guidance at <https://royalsociety.org/journals/authors/author-guidelines/#supplementary-material> to include a suitable title and informative caption. An example of appropriate titling and captioning may be found at https://figshare.com/articles/Table_S2_from_Is_there_a_trade-off_between_peak_performance_and_performance_breadth_across_temperatures_for_aerobic_scope_in_teleost_fishes_/3843624.

Author's Response to Decision Letter for (RSOS-202062.R1)

See Appendix B.

Decision letter (RSOS-202062.R2)

Dear Dr Tsantani,

It is a pleasure to accept your manuscript entitled "The Twenty Item Prosopagnosia Index (PI20) provides meaningful evidence of face recognition impairment" in its current form for publication in Royal Society Open Science. The comments of the reviewer(s) who reviewed your manuscript are included at the foot of this letter.

on behalf of Dr Bruno Rossion (Associate Editor) and Essi Viding (Subject Editor)
openscience@royalsociety.org

Appendix A

The Twenty Item Prosopagnosia Index (PI20) provides meaningful evidence of face recognition impairment

RSOS-202062

Response to Reviewers

Comments to the Authors: Reviewer #1

Tsantani and colleagues present an interesting and timely study about the usefulness of subjective face recognition reports, in particular the PI20, in identifying individuals with objective face recognition abilities. The PI20 has started to become widely adopted by prosopagnosia researchers and I think the manuscript contributes to the debate about whether members of the general population and prosopagnosics have insight into their face recognition abilities. That said, there are some major issues that need to be addressed before recommending publication.

We thank the reviewer for their constructive suggestions.

The first issue is that in the group comparisons (PI20 > 60 vs. 60 or lower) are confounded with how the two samples in the study were recruited, which may be potentially driving some of the differences between the groups. For the prolific sample that provides most of the 'PI20 60 or lower' group, this is more of a general community sample who are coming to the website because they are interested in making money and/or learning about their cognitive abilities. They are also motivated to perform well to keep up their prolific approval rate. In contrast, individuals coming to troublewithfaces.org (who mostly make up the PI20 > 60 group) are likely there because they suspect they have face recognition difficulties. This could potentially bias their self-report questionnaires such as the PI20 and perhaps even their objective test results. Further, individuals going to troublewithfaces.org may also generally have more issues such as undiagnosed autism (it doesn't sound like the authors administered the autism quotient questionnaire), undiagnosed developmental issues/brain injuries, and other potential general visual or memory issues. With such different recruitment for the two sample, I feel like comparisons within each sample would be much more valid.

While we agree with the reviewer's logic, recruiting all the participants from a single source simply wasn't possible; there just aren't enough people on prolific who score above 65 on the PI20 to recruit a large sample via this route.

We accept that the two groups may have differed in their motivation. Crucially, however, we do not believe that any difference in motivation is driving the relationship between high PI20 score and poor performance on the CFMT. If anything, differences in motivation between the groups is likely suppressing the strength of the relationship. We now consider this issue in the revised General Discussion.

Anyone who referenced brain injury or symptoms indicative of a stroke were not eligible to participate. Similarly, anyone who identified as autistic was also ineligible. We have sought to clarify these aspects in the Participants section of the Method.

Interpreting the AQ scores of suspected DPs is not straightforward. The AQ includes a number of items that people with DP are likely to recognize and respond to. For example: I enjoy meeting new people; I find social situations easy; When I'm reading a story, I can easily imagine what the characters might look like; If I try to imagine something, I find it very easy to create a picture in my mind. In light of these items, it is unsurprising that many DPs score higher than typical controls on the AQ; that doesn't mean they have undiagnosed autism.

The other issue is that the cutoff of 60 on the PI20 seems rather arbitrary. Previously it was reported that 65 was a reasonable cutoff to identify prosopagnosics but studies have included participants in prosopagnosia groups with scores of 59. Perhaps a better way to establish a PI20 cutoff is to perform ROC analyses for a particular target population and figure out a reasonable criterion for sensitivity and specificity. From Arizpe et al's recent findings, it doesn't seem like the general population is where the PI20 is most appropriate. It seems like the most useful place for the PI20 is to apply to all individuals that "think they may have prosopagnosia/severe face recognition problems" (or maybe all individuals that visit troublewithfaces.org). Then you could establish a cutoff that would reasonably discriminate those that actually have gold-standard verified prosopagnosia (impaired on at least two face recognition measures) vs. those who do not. These cutoff scores would be quite useful to prosopagnosia researchers who often get contacted by people with face recognition problems and they want to determine if they likely to be severe enough to include in their studies.

The choice of 60 as a cut-off and the lack of justification were issues identified by both reviewers. We apologise for any lack of clarity here.

Gray, Bird and Cook (2017) described PI20 data from 425 typical participants ($M = 41.16$; $SD = 9.93$). The range of "low" PI20 scores (20-60) encompasses the range of scores that fell within 2 SDs of the typical mean in the Gray et al (2017) dataset. In order to be consistent with previous research, however, the high-scoring group now only includes individuals who scored 65 and above. This treatment recognises the fact that scores between 61 and 64 are a little ambiguous: They are more than 2 SDs above the typical mean, but do not reach the (slightly conservative) cut-off of 65 identified by Shah et al (2015). This is now explained in the revised Statistical Procedures section.

The reviewer is correct to note that PI20 scores are not intended to capture the variability in face recognition ability seen within the typical range. People who are slightly below average, average, or slightly above average will respond in very similar ways (e.g., "strongly disagree") to many items (e.g., I sometimes find movies hard to follow because of difficulties recognising characters; Anxiety about face recognition has led me to avoid certain social or professional situations). However, it is still useful to administer the PI20 to people whose ability falls within the normal range: While their responses are relatively uninformative about where they fall within the unimpaired range, their responses do indicate that their face recognition is unimpaired.

Comments to the Authors: Reviewer #2

The research provides a valuable contribution to the literature. It examines the meaningfulness of the PI20 as a tool to identify people who are likely to have Developmental Prosopagnosia. It uses a novel methodology to establish that participants who score high on the PI20 also perform worse on two versions of the CFMT, a widely used objective measure of face recognition ability, than participants who had low scores on the PI20. This indicates that the PI20 could be a useful tool to support screening of potential developmental prosopagnosics. This manuscript will be of particular interest to other researchers in the field who conduct research on DP as identification of larger DP samples is essential to progress our understanding of this developmental disorder. However there are a number of issues that I would hope the authors would address in a revised version of the manuscript.

We thank the reviewer for their constructive suggestions.

Page 4, Line 51-52: can the authors please provide some explanation as to why DP samples have since included participants with scores below the cut-off proposed by Shah et al., (2015) as currently this reads rather contradictory that a cut off was established when the

tool was initially validated and since ignored. What rationale was provided within those papers for including participants not reaching the threshold of 65?

The score of 65 was suggested as a guide, not a hard and fast cut-off whereby people who scored 64 shouldn't be identified as DP. Self-report evidence is informative, particularly when it can be quantified. However, we've always insisted PI20 scores should be used alongside objective evidence from computer-based tasks. In these cases, there was sufficient evidence of impairment on the computer-based tests to include the case in a DP sample.

Our understanding of the measure continues to improve over time. While we still believe 65 is a useful heuristic guide, we've come to see this as a slightly conservative threshold. For example, a score is 61 is more than 2 SDs above the typical mean despite the fact it does not reach this threshold.

Page 5, Line 16 and 17: the authors comment that there has been considerable debate about whether participants have this level of insight into their face recognition abilities – is there any published evidence of this debate that authors could refer readers to for a more detailed discussion of this issue? If not can the authors add a brief summary of this debate. I believe many of the points debated have been included in the introduction section but discussion of these points is rather brief so if readers cannot be directed to a more in depth discussion on this issue the paper would benefit from further detailed discussion of some of these points of debate.

We have provided more references to this debate in the manuscript (paragraph 4 of the Introduction).

Method: were any procedures put in place to ensure the reliability of the online data – beyond ensuring those recruited via prolific had a score of 90%+? (e.g. to ensure participants completed the CFMT tasks individually). If so the MS would benefit from these techniques being detailed to enhance the readers trust in the data.

Online testing is a great innovation that is helping researchers achieve larger sample sizes. However, this approach is associated with some well-known limitations (e.g., inability to control the participants' environment or viewing distance). We now better acknowledge this trade-off in the revised General Discussion.

Page 8, line 29: Can the authors please provide a rationale for using a cut-off point of 61 rather than the 65 originally proposed by Shah et al., (2015)? Some rationale is provided based on the fact other researchers have lowered the cut-off point, but the argument would be stronger with a scientific basis for lowering the cut-off point from that of the initial validation paper.

The choice of 60 as a cut-off and the lack of justification were issues identified by both reviewers. We apologise for any lack of clarity here.

Gray, Bird and Cook (2017) described PI20 data from 425 typical participants ($M = 41.16$; $SD = 9.93$). The range of "low" PI20 scores (20-60) encompasses the range of scores that fell within 2 SDs of the typical mean in the Gray et al (2017) dataset. In order to be consistent with previous research, however, the high-scoring group now only includes individuals who scored 65 and above. This treatment recognises the fact that scores between 61 and 64 are a little ambiguous: They are more than 2 SDs above the typical mean, but do not reach the (slightly conservative) cut-off of 65 identified by Shah et al (2015). This is explained in the revised Statistical Procedures section.

More generally, the aim of the paper is not entirely clear. The explicitly stated aim is to address the question of whether DPs and controls have sufficient insight in their face recognition ability for the PI20 to be meaningful, a question that I believe the manuscript sufficiently addresses. However, the introduction appears to aim to provide a defence of the PI20 as a tool to identify people with DP. The first aim is achieved but I am not convinced of the latter. If the intended use of the PI20 is to identify suspected DPs it seems unusual that the authors did not conduct a full battery assessment of the high scorers to establish whether their performance is consistent with the profile of DP. That is, yes the authors have found that high PI20 scorers perform worse on the CFMT, a frequently used test that forms part of a typical assessment for DP. However, a poor score on CFMT alone or combined with a poor score on the CFMT-a is not sufficient to identify a person as having DP. Further tests / assessment is required to rule out alternative explanations of the face recognition difficulties (e.g. low-level visual difficulties, more general object recognition difficulties). These additional assessments would have better established the PI20's ability to discriminate between DPs and non-DPs. If the authors don't have data available to add this level of detail to the manuscript some consideration of this limitation of the current study is warranted. The findings provide convincing evidence that there is an association between PI20 scores and performance on CMFT and CMFTa as objective measures of face recognition but that doesn't necessarily mean an association with DP per se which I think needs to be considered.

The aim of the current paper is the former (to determine whether DPs and typical controls have sufficient insight in their face recognition ability for the PI20 to be meaningful). Importantly, however, the interest in this question arises from the use of the PI20 scale in the identification of cases of DP.

At no point do we refer to the high-PI20 scorers as "Developmental Prosopagnosics" even where individuals score poorly on both versions of the CFMT.

It is widely accepted that some DPs have co-occurring object-recognition difficulties (see Geskin & Behrmann, 2018, *Cognitive Neuropsychology*). However, we wholeheartedly endorse the reviewer's point that the contribution of low- and mid-level visual differences to individual differences in face recognition ability, remain poorly understood. We now reference this suggestion in the General Discussion.

Appendix B

The Twenty Item Prosopagnosia Index (PI20) provides meaningful evidence of face recognition impairment

RSOS-202062

Response to reviewers

Comments to the authors: Reviewer #1

While I appreciate the authors' changes, the issue of having a confound that participants who comprised the low PI-20 group vs. high PI-20 were recruited from different sources remains unresolved. Maybe we just are learning that individuals who self-select and go to troublewithfaces.com are generally worse at face recognition than the general population of individuals at prolific. I'm not sure if more can be concluded from the results.

While we disagree with the reviewer, we thank them for their constructive comments and for their efforts on our behalf.

Ideally all participants would be recruited from the same source – certainly, this would be more elegant. In practice, however, this is extremely difficult to implement because of the low prevalence of DP in the general population. For example, in their sample (N = 425), Arzipe et al identified only 12 people who scored in the DP range on the PI20. In our view, having such small numbers of individuals who score within the suspected DP range is far more likely to lead to misleading conclusions, and should be seen as the more pressing concern.

The fact that people approach websites such as troublewithfaces.org implies that they think they may have poor face recognition. We don't see that this is a confound; rather the aim of the study is to quantify and test this intuition. We haven't tried to hide the fact that these people think they have face recognition problems – on the contrary, this forms the basis of our key independent variable (i.e., their PI20 scores).

As we explain in the discussion, there may be a confound in terms of participants' motivation. The high-scorers recruited through troublewithfaces.org have typically sought out the research team without any expectation of financial reimbursement. They are invested in the research question in a way that participants recruited via prolific may not be. However, if differential motivation has any effect, we believe it leads to the *underestimation* of the diagnostic performance of the PI20 (e.g., poorly motivated controls score lower on the CFMT than might be expected based on their PI20 scores). In other words, differential motivation is unlikely to explain our results.

At present, the principal users of the PI20 are DP researchers who are trying to diagnose DPs for inclusion in research samples. In the overwhelming majority of cases, suspected DPs approach researchers via specialist websites such as troublewithfaces.com and faceblind.org. The way we recruited high-scorers in the present study, thus closely reflects how the PI20 is used in research settings. Our results imply that the scores obtained by researchers can aid diagnostic decisions.

Also, I would have preferred the authors perform an ROC analysis similar to Arzipe et al rather than go back to a relatively arbitrary cutoff score.

We have now conducted a ROC analysis to determine the optimal PI20 cut-off score. The PI20 cut-offs identified by this (post-hoc) ROC analysis were 60.5 for participant splits based on diagnostic CFMT cut-offs of 65% and 60%, and 75.5 for the split based on the strictest CFMT diagnostic cut-off (55%). We thank the reviewer for suggesting this excellent addition.

We agree that the choice of 65 as a cut-off by Shah et al. was somewhat arbitrary. However, rightly or wrongly, 65 is the cut-off that was suggested by the authors and the one that has been used since. We believe most researchers would therefore agree that examining the effectiveness of this cut-off is a useful contribution to the literature.

Because our aim was to compare CFMT performance across low-PI20 scorers and high-PI20 scorers, selecting an optimal PI20 cut-off based on CFMT performance and then using that cut-off to divide our groups and compare CFMT performance would result in a circular analysis.

Comments to the authors: Reviewer #2

This revised manuscript clearly addresses the major concerns raised by the reviewers, in particular the issue relating to the cut-off point for the high PI20 group. I welcome the revised analysis using the more conservative cut-off of 65. Whilst both the introduction and discussion sections remain brief I believe this article does provide a valuable contribution to the field, serving to illustrate the insight participants have into their own face recognition ability which further establishes the PI20 as a useful tool to support the identification of participants with suspected DP. I would therefore recommend acceptance of the article with just a couple of minor revisions;

We thank the reviewer for their constructive comments.

Page 8 - the new paragraph says "this was the threshold originally identified Shah et al [23]" - I believe "by" is missing after identified within this sentence.

Thanks for spotting this - we have now corrected this typo.

Page 10 - final sentence about low-level visual capacities. Could the authors please expand on this point to clearly explain why this is important for the readers to consider i.e. the potential impact it could have on the findings. It is now clear that participants were excluded if they had autism, schizophrenia or other mental health conditions etc. This additional information is the method is very welcome. However, it's not clear any attempt was made to check that poor performance on all measures wasn't due to differences in low-level visual perception / ability. It's possible someone with poor vision / low-level visual perceptual difficulties might score high on the PI20 and low on the CFMT but that the origins of their impairment be different to that of someone with DP who has scored high on the PI20 and low on the CFMT. So this point needs to be considered more fully not just within the context of online testing which is where it is currently raised.

We agree with the reviewer that not measuring low-level visual capacities is a limitation of our study. However, the fact that the study was conducted online is highly relevant as this precludes meaningful tests of low-level vision.

We have added a paragraph to the discussion in which we address the question of whether the association seen between scores on the PI20 and CFMT might be driven by the presence of people with poor visual acuity.

The contribution of low-level visual problems to PI20 responses and poor face recognition is certainly an important avenue for future research. However, previous findings suggest that low-level visual deficits are unlikely to be responsible for the association between PI20 scores and CFMT performance seen here. If the PI20 captured variability in visual acuity, one would expect PI20 scores to be a good

predictor of performance on non-face object recognition tasks. However, this is not the case (Shah et al, 2015; Biotti et al., 2017; Gray et al., 2019).

Participants' responses on the PI20 may be informed by a lifetime of experience (e.g., When I was at school, I struggled to recognize my classmates; It is easy for me to recognise individuals in situations that require people to wear similar clothes; It is hard for me to recognize familiar people when I meet them out of context), and therefore relatively robust to recent deterioration in visual acuity.

Discussion - the authors might want to consider adding a conclusion paragraph at the end of the ms. The additions to the discussion are an improvement to the ms addressing some of the concerns raised by the reviewers of the original ms but they feel somewhat bolted on at the end of the ms and a return to the take home message, in spite of the limitations now acknowledged, would be beneficial to end the ms focussed on the important study aims.

We thank the reviewer for this suggestion. We have now added a conclusion paragraph.

Comments to the authors: Reviewer #3

The paper presents a systematic evaluation of the diagnostic value of the PI-20 - a self-report measure of face recognition difficulties in daily life - in predicting scores on a test of relatively short-term unfamiliar face memory - the CFMT - which is a standard objective measure of face recognition ability. This is a useful study and method, presentation of results and reporting is generally high quality. The paper is well written.

We thank the reviewer for his constructive comments.

My major concern is the method for separating Low v High PI-20 scorers. The authors discard an undisclosed* proportion of participants with PI-20 scores between 61 and 64 , terming this data as 'ambiguous' . But I don't think it is acceptable to present this incomplete picture in a diagnostic evaluation like this -- because the whole point of a diagnostic tool is that it allows researchers to set a criteria for diagnosis. It is perfectly OK to assess different criteria -- for example comparing the diagnostic value of a 60 v 65 PI-20 cutoff , but to exclude bands of data from the analysis is misleading and will lead to an inflated sense of the diagnostic value in researchers / clinicians that read the paper by casting their eye over the abstract and the main data figures. This practice is equivalent to conducting a psychophysics experiment, where participants provide judgments of perceptual certainty on a likert scale, and removing data around the midpoint of the scale before calculating a participants' accuracy. At the very least the CFMT data from this ambiguous group should be presented in the figures.

To avoid any ambiguity we have now added the data from 9 participants who scored between 61-64 on the PI20 to our group of low-PI20 scorers, and re-run all analyses. The pattern of results has remained the same.

But perhaps there is an opportunity to do something more holistic. For example, Area Under the ROC curve is a useful standard approach used in medical diagnosis (e.g. <https://www.sciencedirect.com/science/article/pii/S1556086415306043> , <https://www.ncbi.nlm.nih.gov/pmc/articles/PMC3755824/>). In the present study, this could be used by classifying the participants as CP / nonCP based in the CFMT, and using the continuous PI-20 score to predict this state (perhaps repeating this analysis for the different CFMT cutoffs in the current analysis).

We have conducted a ROC analysis for each of our three CFMT cut-offs (55, 60, and 65%). The PI20 showed better-than-chance ability to discriminate between CFMT-

diagnosed DPs and non-DPs, with AUC values ranging between 75% and 88%. Based on the ROC curves, we also estimated the optimal PI20 cut-offs for each CFMT cut-off. The method (p7) and results (p9) of these analyses are described in the revision. We thank the reviewer for suggesting this excellent addition.

The proportion of participants with PI-20 between 61 and 64 is undisclosed because the total number of participants reported to have completed the experiment do not include these participants. But given that all participants completed the PI-20, I am assuming that the full dataset must have included participants scoring in this range.

We apologise for any confusion. We have clarified this aspect in the revision (p11).

Figure 1 - (a) and (b) -- I think this plot would be clearer if y-axis was labelled 'Percent correct'. Also - could the two figures be combined simply by adding the cutoff lines in (b) to (a)? Please also clarify in the figure captions that the cutoffs relate to CFMT scores. I know it can be deduced from the information but worth making it abundantly clear.

The figures have been revised along the lines suggested.

Comments to the Authors: Reviewer #4

Having read the authors' response to reviews as well as the revised text, I have found them to be responsive to the main issues raised in the last round of review. I agree that their justification of the cut-off values used to identify subgroups of suspected DP and non-DP participants are more clearly stated and I think that while recruiting from different outlets is not ideal, it also seems like it was likely necessary to ensure a large enough sample with sufficient variability to carry out their analysis.

We thank the reviewer for their constructive comments.

I think the only thing that could strengthen the main results somewhat is highlighting how the correlation between PI20 scores and CFMT performance plays out in both subgroups: I imagine poorly, which is largely their point! I think this could provide useful context given the conflicting results they describe in the introduction, but would not want to make this a condition for acceptance.

We now report the correlation between PI20 scores and CFMT performance (for CFMT-original and CFMT-A) for our low-PI20 scorers and for our high-PI20 scorers. As expected, the correlations are low, ranging from -.21 to -.36. This confirms that the PI20 has limited ability to make fine-grained distinctions between individuals who have slightly better/worse face recognition than each other, and is better suited for making broad distinctions between potential DPs and individuals with typical face recognition ability.

Otherwise, I think that the potential comorbidities of suspected DP with other aspects of object recognition and lower level visual processes could be given a little more space, but these also seem somewhat ancillary to the main point: The PI20 does indeed support the identification of participant groups with low vs. high face recognition performance as indexed by the CFMT. While more generality (perhaps examining instruments like the CFPT or the Glasgow Face Matching test) could be additionally useful, the paper accomplishes the narrower goal highlighted in the introduction.

We have developed our discussion of the potential presence of individuals with undiagnosed visual difficulties in our sample (p12).